# Single-cell profiling identifies *IL1B^hi* macrophages associated with inflammation in PD-1 inhibitor-induced inflammatory arthritis

Ziyue Zhou [1,2,3,11], Xiaoxiang Zhou [4,11], Xu Jiang [3,5,11], Bo Yang[6,11], Xin Lu[6], Yunyun Fei[1,2,3], Lidan Zhao[1,2,3], Hua Chen[1,2,3], Li Zhang[7], Xiaoyan Si[7], Naixin Liang[8], Yadong Wang[8], Dan Yang [1,2,3], Yezi Peng[1,2,3], Yiying Yang[1,2,3], Zhuoran Yao[9], Yangzhige He[3,5], Xunyao Wu [3], Wen Zhang[1,2,3], Min Wang[10], Huaxia Yang [1,2,3] ✉ & Xuan Zhang [10] ✉

Inflammatory arthritis (IA) is a common rheumatic adverse event following immune checkpoint inhibitors treatment. The clinical disparities between IA and rheumatoid arthritis (RA) imply disease heterogeneity and distinct mechanisms, which remain elusive. Here, we profile CD45+ cells from the peripheral blood or synovial fluid (SF) of patients with PD-1-induced IA (PD-1-IA) or RA using single-cell RNA sequencing. We report the predominant expansion of *IL1B^hi* myeloid cells with enhanced NLRP3 inflammasome activity, in both the SF and peripheral blood of PD-1-IA, but not RA. *IL1B^hi* macrophages in the SF of PD-1-IA shared similar inflammatory signatures and might originate from peripheral *IL1B^hi* monocytes. Exhausted CD8+ T cells (Texs) significantly accumulated in the SF of patients with PD-1-IA. *IL1B^hi* myeloid cells communicated with CD8+ Texs possibly via the CCR1-CCL5/CCL3 and CXCL10-CXCR3 axes. Collectively, these results demonstrate different cellular and molecular pathways in PD-1-IA and RA and highlight *IL1B^hi* macrophages as a possible therapeutic target in PD-1-IA.

Programmed cell death 1 (PD-1) inhibitors have revolutionized the treatment of cancer. By releasing the brake established by this immune checkpoint, PD-1 inhibitors activate antitumor immunity and thus have achieved tremendous success in clinical trials of a growing number of cancer types. However, excessive activation of the immune system by PD-1 inhibitors sometimes causes damage to host tissues, leading to a spectrum of immune-related adverse events (irAEs)[1].

Among irAEs, rheumatic irAEs are being increasingly recognized in clinical practice[2]. The most common rheumatic irAE appears to be arthralgia, occurring in 11.4–13.6% of patients receiving immune checkpoint inhibitor (ICI) combination therapies[3]. Specifically, inflammatory arthritis (IA) was found to persist in approximately half of patients after ICI cessation, with a median follow-up time of 9 months[4]. Current guidelines for the management of PD-1 inhibitor-induced inflammatory arthritis (PD-1-IA) are mainly based on expert opinions[5,6]. While the current recommendations for steroids and immunosuppression are efficient in most cases, the long-term medication generally needed in PD-1-IA raises particular concerns about compromising antitumor immunity. Indeed, both high-dose glucocorticoids and second-line immunosuppression for irAEs have been reported to impair ICI efficacy in patients with melanoma[7,8]. Therefore, the pathogenesis of PD-1-IA urgently needs to be elucidated to guide the treatment algorithm without impeding antitumor efficacy.

**Fig. 1 | Overview of CD45+ immune cells in inflammatory arthritis and controls. a** Workflow showing the collection and processing of samples for scRNA-seq, flow cytometry, and ELISA. **b** UMAP projection of 9 major cell types in synovial fluid mononuclear cells (SFMCs) (*n* = 3 individuals per group). **c** The proportions of major cell types in SFMCs from all individuals in each patient group (IA_act and seropositive RA). **d** UMAP projection of 9 major cell types in peripheral blood cells (PBMCs) (*n* = 3–4 individuals per group). **e** The proportions of major cell types in PBMCs from all individuals in each patient group (IA_act, IA_rem, seropositive RA, and HC). IA_act active inflammatory arthritis, IA_rem inflammatory arthritis in remission, RA rheumatoid arthritis, HC healthy control. MΦ macrophages, T.cycl. cycling T cells.

Our knowledge about the mechanism of PD-1-IA is still limited. Although clinically resembling rheumatoid arthritis (RA) in terms of joint distribution and joint effusion, PD-1-IA is generally seronegative for anti-citrullinated protein antibodies or rheumatoid factor, suggesting that different mechanisms underlie the two disorders[2]. One case reported extensive T-cell infiltration in the synovial fluid (SF) of nivolumab-induced IA, which was similar to the features of one of the three RA controls. However, B cells were undetectable in the nivolumab-induced IA case, which was different from the features of the RA controls[9]. A recent study suggested that peripheral CD8+ T cells in ICI-induced IA patients had a distinct immune-effector profile compared with those in ICI-treated patients without irAEs but that this profile overlapped with the CD8+ T-cell profile of patients with RA[10]. Overall, PD-1-IA seems to be different from RA while still sharing some

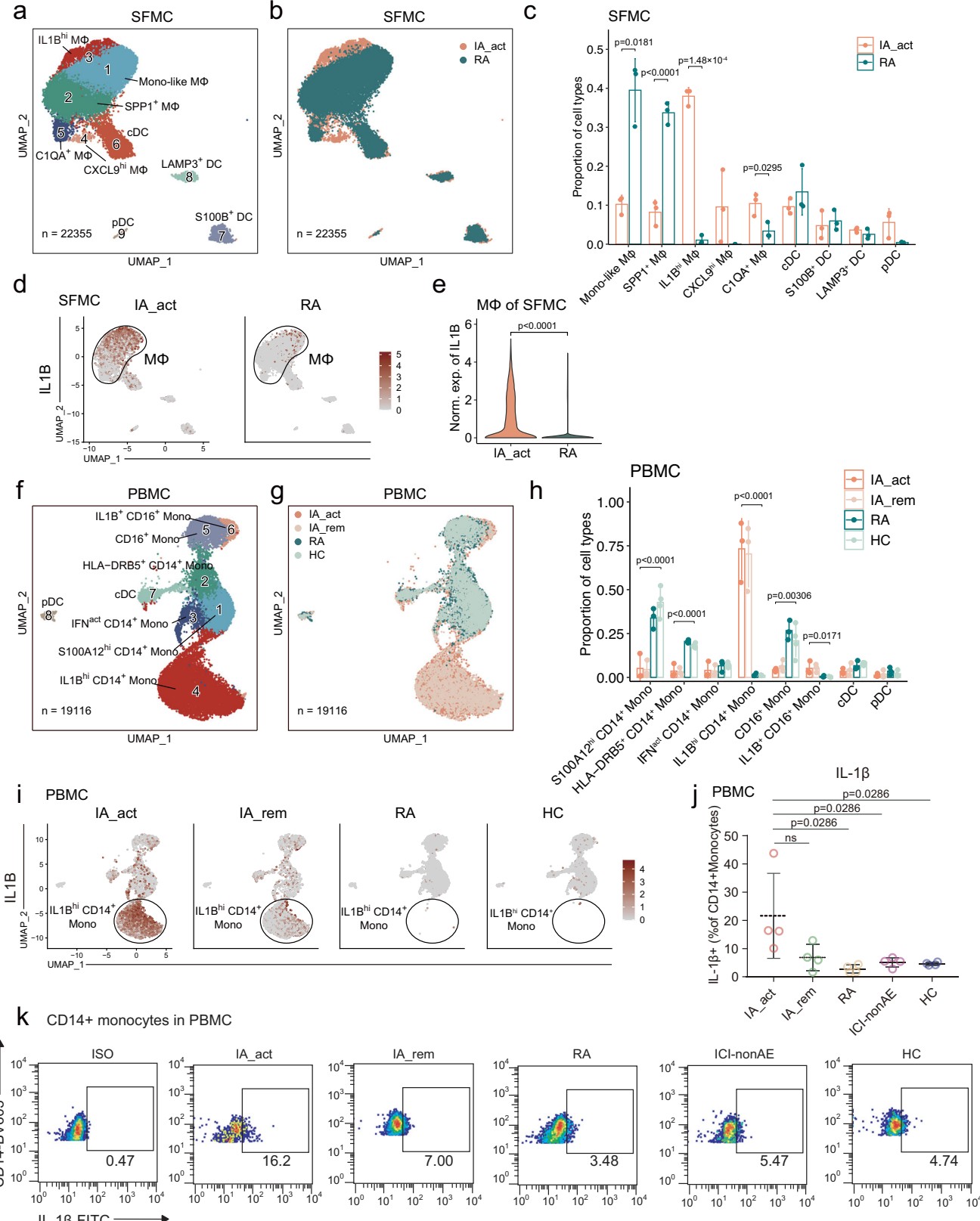

common molecular features. Nonetheless, the pathogenic immune response in PD-1-IA and whether a shared pathogenic immune response exists between PD-1-IA and RA remain unknown.

In this study, we aim to comprehensively characterize the immunophenotype of SF and peripheral blood of patients with active

PD-1-IA (IA_act), patients with PD-1-IA in remission (IA_rem), patients with seropositive RA and healthy controls (HCs) using single-cell RNA sequencing (scRNA-seq). We identify *IL1B*[hi] macrophages and exhausted CD8[+] T cells (CD8[+] Texs) as key cellular features of PD-1-IA. Furthermore, we reveal that the crosstalk between *IL1B*[hi] macrophages and

**Fig. 2 | Identification of IL1B^hi macrophages/monocytes as IA-associated myeloid cells. a** Identification of 9 subclusters of myeloid cells across all SFMC samples. **b** Distribution of myeloid cell subclusters of SFMCs between IA_act and RA. **c** Barplot showing the proportion (mean ± SD) of each myeloid cell subcluster in IA_act and RA with *p*-values calculated by unpaired two-sided *t*-tests. **d** UMAP plots showing *IL1B* gene expression in myeloid cells in SFMCs from IA_act (left) or RA (right) patients. **e** Quantification of the differences in *IL1B* gene expression of macrophages of SFMCs between the IA_act and RA by a two-sided Wilcoxon test. **f** Identification of 8 subclusters of myeloid cells across all PBMC samples. **g** Distribution of myeloid cell subclusters in PBMCs among patient groups (IA_act, IA_rem, RA, and HC). **h** Barplot showing the proportion (mean ± SD) of each myeloid subcluster among the patient groups with one-way two-sided ANOVA tests. **i** UMAP plots showing the *IL1B* gene expression of myeloid cells in PBMCs among the patient groups. **j** Quantification of IL1β^+ CD14^+ monocytes (percentage of IL1β^+ cells in CD14^+monocytes, mean ± SD) among the patient groups by flow cytometry. The data show *n* = 4 biological replicates over three independent experiments. Paired two-sided *t*-test compared the IA_act group with the IA_rem group. Wilcoxon tests compared the IA_act group with the RA, ICI-nonAE, or HC group. ns, nonsignificant. **k** Representative flow cytometry plots for (**j**). ICI-nonAE ICI-treated patients without irAEs, IA_act active inflammatory arthritis, IA_rem inflammatory arthritis in remission, RA rheumatoid arthritis, HC healthy control, ISO isotype, Mono-like monocyte-like, MΦ macrophages.

CD8^+ Texs might promote the pathogenesis of PD-1-IA. Thus, this work provides clues about potential therapeutic targets in PD-1-IA.

## Results

### The immunological landscapes of SFMCs and/or PBMCs of PD-1-IA patients, RA patients and HCs

We comprehensively analyzed CD45^+ immune cells in the synovial fluid and peripheral blood of patients with IA_act (*n* = 5) and active seropositive RA (*n* = 8) using scRNA-seq, flow cytometry, and ELISA. Additionally, CD45^+ immune cells were obtained from the paired peripheral blood of IA_rem patients (*n* = 5) and HCs (*n* = 8) (Fig. 1a). We also retrieved external scRNA-seq dataset of PBMCs from the patients who received ICI treatment but did not develop irAEs (*n* = 7) from a published article as additional control[11]. The median age of PD-1-IA patients at IA onset was 59 years (IQR = 51–62 years), with female predominance (female:male = 4:1). The median disease duration between anti-PD-1 treatment and the onset of arthritis was 97 days (IQR = 70–269 days). Three patients had polyarthritis involving both large and small joints, whilst two patients had oligoarthritis only affecting the bilateral knee joints. The demographics, clinical characteristics, and treatments are listed in the supplementary materials (Supplementary Tables 1, 2).

To investigate the immune cell profile of inflamed joints, we performed scRNA-seq of CD45^+ SFMCs in IA patients (*n* = 3) and seropositive RA patients (*n* = 3) (Fig. 1a). The combined SFMC scRNA-seq dataset contained 40482 high-quality cells from well-defined immune lineages. To analyze the systemic immunologic responses in PD-1-IA, we performed scRNA-seq of sorted CD45^+ PBMCs from IA_act (*n* = 3) and IA_rem (*n* = 3) patients and compared the results with data for the same cells from RA patients (*n* = 3) and HCs (*n* = 4). The combined PBMC scRNA-seq dataset contained 64,819 high-quality cells from well-defined immune lineages.

Based on unsupervised clustering of SFMCs, 9 major clusters including T cells, cycling T cells, natural killer (NK) cells, macrophages, B cells, conventional dendritic cells (cDCs), plasmacytoid dendritic cells (pDCs), S100B^+ dendritic cells (DCs), and LAMP3^+ DCs were defined by canonical markers (Fig. 1b, Supplementary Fig. 1a). Striking disparities in SFMCs were observed between IA_act and RA patients. Compared with RA, IA_act showed a significant increase in the proportion of T cells and significant decreases in the proportions of macrophages, cDCs, and S100B DCs in SFMCs (Fig. 1c, d; Supplementary Fig. 1b). The actual counts and flow cytometry of the cell subsets in each sample were shown in Supplementary Fig. 1c, d. Similarly, a total of 9 major cell lineages including T cells, cycling T cells, NK cells, monocytes, B cells, cDCs, pDCs, plasma cells, and platelets were identified by canonical markers in PBMCs (Fig. 1d, Supplementary Fig. 1e). The proportions of major cell lineages, except for that of platelets, did not significantly differ among the four groups (Fig. 1e, Supplementary Fig. 1f). These results showed that IA_act was associated with major changes in T cells and myeloid cells in the synovial fluid but not in the peripheral blood.

### Expanded *IL1B*^hi myeloid cells in PD-1-IA synovial fluid and peripheral blood

To identify PD-1-IA-related cell clusters, we performed unsupervised clustering of myeloid cells. Based on canonical myeloid markers (Supplementary Fig. 2a), 9 subclusters of myeloid cells were defined in SFMCs; these subclusters included 5 monocyte/macrophage subclusters (monocyte-like macrophages, *SPP1*^+ macrophages, *IL1B*^hi macrophages, *CXCL9*^hi macrophages, and *C1QA*^+ macrophages) and 4 DC subclusters (cDCs, *S100B*^+ DCs, *LAMP3*^+ DCs, and pDCs) (Fig. 2a). The IA_act group was characterized by a predominant proportion of *IL1B*^hi macrophages, which was almost absent in the RA groups (Fig. 2a–e, Supplementary Figs. 2b, 3a–f). In contrast, the RA groups were significantly enriched in monocyte-like macrophages and *SPP1*^+ macrophages (Fig. 2c). Furthermore, we performed flow cytometry and observed an increased proportion of IL1β^+ CD11b^+ synovial macrophages from one IA_act patient compared to the one RA patient (Supplementary Fig. 2c, gating strategy in Supplementary Fig. 13a).

In PBMCs, 8 subclusters of myeloid cells were defined according to canonical myeloid markers (Supplementary Fig. 2d); these subclusters included 6 monocyte subclusters (*S100A12*^hi CD14^+ monocytes, *HLA-DRB5*^+ CD14^+ monocytes, IFN-activated [IFN^act] CD14^+ monocytes, *IL1B*^hi CD14^+ monocytes, CD16^+ monocytes, and *IL1B*^+ CD16^+ monocytes) and 2 DC subclusters (cDCs and pDCs) (Fig. 2f). Notably, we also observed that the two *IL1B*^hi subclusters (*IL1B*^hi CD14^+ monocytes and *IL1B*^+ CD16^+ monocytes) were nearly exclusive to the IA_act and IA_rem groups (Fig. 2g, h, Supplementary Fig. 2e, Supplementary Fig. 3g–k). Intriguingly, the expression of *IL1B* was dramatically decreased in the IA_rem group compared with the IA_act group (Fig. 2i). This finding was further validated by flow cytometry (Supplementary Fig.2c, gating strategy in Supplementary Fig. 13a). The fraction of IL1β^+ CD14^+ monocytes was significantly higher in the IA_act group than in the RA group, ICI-nonAE group and HC group (Fig. 2j, k, gating strategy in Supplementary Fig. 13b). Overall, we found that *IL1B*^hi myeloid cells were significantly enriched in both the local joint and peripheral blood of PD-1-IA patients, suggesting a pathogenic role for *IL1B*^hi myeloid cells in PD-1-IA.

### *IL1B*^hi macrophages in SFMCs share similar inflammatory signatures with *IL1B*^hi monocytes in PBMCs and may differentiate from PBMCs

To identify the molecular pathways and potential treatment targets in PD-1-IA, we investigated the gene expression features of IA-associated myeloid cell subsets: *IL1B*^hi macrophages in SFMCs and *IL1B*^hi monocytes in PBMCs. In SFMCs, the top 10 DEGs between *IL1B*^hi macrophages and other monocyte/macrophage subsets were chemokines (*CXCL10*, *CCL3L1*, *CCL8*), interferon-inducible genes (*IFIT2*, *IRF1*, *GBP1*, *GBP5*) and genes related to the M1 macrophage phenotype (*APOBEC3A*[12], *TNFSF10*[13], *CALHM6*[14]) (Fig. 3a). We further performed gene set enrichment analysis (GSEA) with hallmark pathways from MSigDB[15]. Multiple inflammatory pathways were significantly upregulated in *IL1B*^hi macrophages and were all designated PD-1-IA-related inflammatory pathways: IFNγ response, IFNα response, TNF signaling through NFκB,

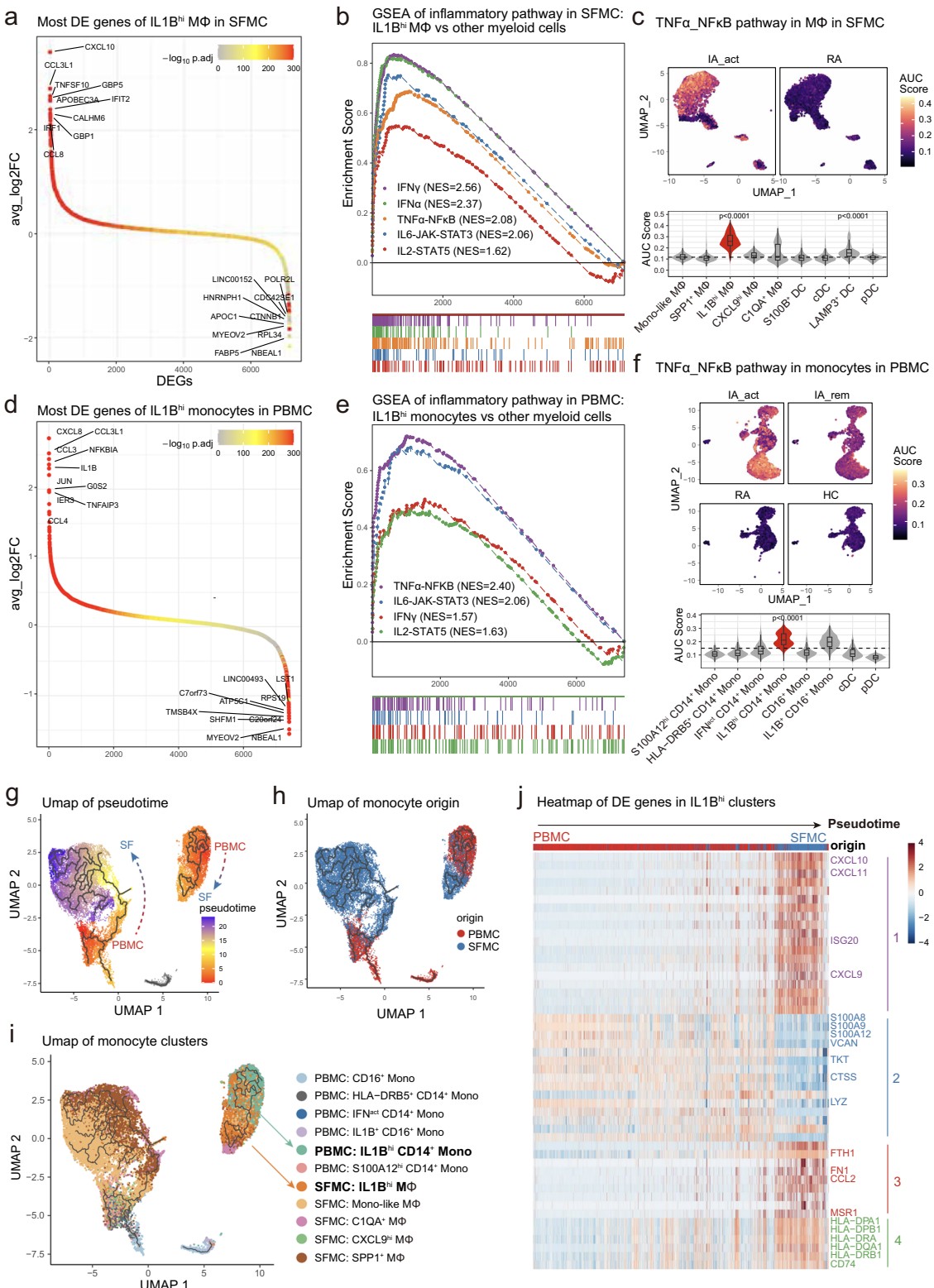

IL6-JAK-STAT3 and IL2-STAT5 pathways (Fig. 3b, Supplementary Fig. 4a). Direct comparison of synovial macrophages between the IA_act and RA groups confirmed the upregulation of IA-related inflammatory pathways in IA_act (Supplementary Fig. 4b, 5a, 5b). We investigated the expression of the IA-related pathway at the single-cell level by calculating the AUC scores for each pathway in each cell and further visualized AUC scores with a dimplot. The *IL1B^hi* macrophage cluster in the SFMCs of IA_act patients significantly upregulated all IA-

related inflammatory pathways compared with the other clusters, which indicated the proinflammatory pathogenic roles of *IL1B^hi* clusters (Fig. 4c, Supplementary Fig. 6a–d).

Similar to the results for SFMCs, *IL1B^hi* monocytes in the peripheral blood also demonstrated proinflammatory features. The most significant DEGs of *IL1B^hi* monocytes in PBMCs were chemokines (*CXCL8, CCL3L1, IL1B, CCL4*) (Fig. 3d, Supplementary Fig. 7a). GSEA showed that the IA-associated inflammatory pathways were also

**Fig. 3 | *IL1B^hi* macrophages in SFMCs were marked with an inflammatory signature and may differentiate from *IL1B^hi* monocytes in PBMCs. a** Differentially expressed genes between *IL1B^hi* macrophages and other myeloid subclusters in SFMCs, with the top 10 and bottom 10 differentially expressed genes labeled. **b** Gene set enrichment analysis of the *IL1B^hi* macrophage subcluster versus other myeloid subclusters in SFMCs for inflammatory pathways. **c** UMAP visualization (top) and violin and box plot (bottom) showing the TNF_NFκB pathway signature score among myeloid subclusters of SFMCs and between IA_act and RA. The signature score was calculated by the AUCell algorithm (see the "Methods" section). **d** Differentially expressed genes between the *IL1B^hi* monocyte subcluster and other myeloid subclusters in PBMCs, with the top 10 and bottom 10 differentially expressed genes labeled. **e** Gene set enrichment analysis of *IL1B^hi* monocytes compared with other myeloid subclusters in PBMCs for inflammatory pathways. **f** UMAP visualization (top) and violin and box plot (bottom) showing the differences in the TNF_NFκB pathway signature score among myeloid subclusters of PBMCs and patient groups. The signature score was calculated by the AUCell

algorithm (see Methods). **g** Integration of macrophages from SFMCs and monocytes from PBMCs colored by pseudotime. Trajectories are indicated by black lines, developing from low to high pseudotime values. **h** UMAP distribution of the origin of monocytes/macrophages. **i** UMAP distribution of subclusters of monocytes/macrophages. The *IL1B^hi* clusters on the interested trajectory were marked. **j** Heatmap of the top 50 pseudotime differentially expressed genes of 4 modules across the *IL1B^hi* CD14^+ monocyte-to-*IL1B^hi* macrophage trajectory. In the box-plots of the bottom panel of (**c**) (*n* = 6) and (**f**) (*n* = 13), the center lines of the boxes denote the median of the AUC score; the lower and upper limits of the boxes denote the 25% and 75% quantile, respectively. Two-sided Wilcoxon tests comparing the median AUC score of one subcluster with all other clusters. The horizontal dashed line denotes the median AUC score of all the cell clusters, and only the clusters with significantly increased medians were denoted with *p*-value. IA_act active inflammatory arthritis, IA_rem inflammatory arthritis in remission, RA rheumatoid arthritis, HC healthy control, Mono-like monocyte-like, MΦ macrophages.

significantly upregulated in *IL1B^hi* monocytes and included IFNγ response, TNF signaling through NFκB, IL6-JAK-STAT3 and IL2-STAT5 pathways, among which the TNF signaling pathway ranked above the others (Fig. 3e; Supplementary Figs. 4c, 6e–g, 7b). Further analysis of pathway enrichment also showed that the TNF signaling pathway was the most prominently upregulated pathway in the IA_act group compared with the disease remission, RA and HC groups (Supplementary Fig. 4d–f). Among all the cell clusters, the *IL1B^hi* CD14^+ monocyte cluster exhibited the strongest upregulation of the TNF signaling pathway (Fig. 3f). All IA-associated pathways were significantly upregulated in the IA_act group compared with the RA and HC groups, while inflammatory pathways other than TNF signaling did not differ between active disease and remission (Fig. 3f, Supplementary Fig. 8a–f). Therefore, only the TNF signaling pathway was related to disease activity and possibly the core pathogenic pathway of PD-1-IA. Other upregulated proinflammatory pathways (IFNγ, IFNα, IL6, and IL2) may reflect the systemic immune response induced by ICI administration.

We further analyzed the gene expression kinetics of monocytes and macrophages using the Monocle3 package, an unsupervised algorithm[16–18]. We reconstructed two trajectories from the combined single-cell dataset of monocytes and macrophages from SFMCs and PBMCs (Fig. 3g, h). Notably, *IL1B^hi* monocytes and macrophages formed a trajectory distinct from other clusters, indicating a close cell transitional order (Fig. 3i). Ordered by pseudotime (an abstract unit of cell transition), the DEGs along the *IL1B^hi* cell trajectory clustered into 4 distinct modules. Module 1 was mainly composed of proinflammatory genes, while module 4 was mainly composed of marker genes of macrophages. The *IL1B^hi* cell trajectory reflected the transition from peripheral blood monocytes to synovial fluid proinflammatory macrophages, whose phenotype resembled that of M1 macrophages (Fig. 3j). Based on the genetic dynamics, we assumed that *IL1B^hi* macrophages may differentiate from peripheral *IL1B^hi* monocytes.

### Elevated NLRP3 inflammasome pathway activity in *IL1B^hi* myeloid cells from either PBMCs or SFMCs

IL1β is classically produced following the activation of the NLRP3 inflammasome[19]. To identify the mechanisms of *IL1B^hi* myeloid cell expansion in PD-1-IA, we examined the NLRP3 inflammasome pathway in *IL1B^hi* myeloid cells by assessing the signatures of involved genes. In addition to utilizing signatures derived from the GO database and KEGG database[15], we generated a customized signature that included only the core genes related to NLRP3 inflammasome activation (*NLRP3, PYCARD, CASP1, IL1B, IL18, GSDMD*). In SFMCs, the activities of the three signatures were mostly elevated in *IL1B^hi* macrophages and were significantly enriched in the IA_act group compared with the RA group (Fig. 4a, b; Supplementary Fig. 9a, b). The coexpression of the *NLRP3* gene and *IL1B* gene, as well as the coexpression of NLRP3

inflammasome core pathway components and the *IL1B* gene, was significant in single-cell transcriptomics (Fig. 4c, Supplementary Fig. 9c).

Similarly, in PBMCs, the activities of NLRP3 inflammasome signatures were mostly elevated in *IL1B^hi* monocytes and were significantly enriched in the IA group compared with the RA group and HC group (Fig. 4d, e; Supplementary Fig. 9a, b). Furthermore, the activities of NLRP3 inflammasome signatures, except for the NLRP3 inflammasome complex assembly signature, were elevated in the IA_act group compared with the IA_rem group (Fig. 4d). The coexpression of the *NLRP3* gene and *IL1B* gene, as well as the coexpression of NLRP3 inflammasome core pathway components and the *IL1B* gene, was significant in single-cell transcriptomics (Fig. 4f, Supplementary Fig. 9d). Together, these results indicated the important role of the NLRP3 inflammasome pathway in *IL1B^hi* myeloid cells and its potential as a therapeutic target in PD-1-IA.

### Increase in the CD8^+ Tex population in SFMCs of PD-1-IA patients

Next, we examined the synovial T-cell and NK cell compartments in detail. Among the 12 well-defined T/NK subclusters, 5 were CD4^+ T cell subclusters (CD4^+ central memory T cells [Tcms], *SELL*^+ CD4^+ Tcms, CD4^+ effector memory T cells [Tems], CD4^+ Texs, and CD4^+ regulatory T cells [Tregs]), 5 were CD8^+ T cell subclusters (CD8^+ mucosal-associated invariant T cells [MAITs], CD8^+ Tems, CD8^+ cytotoxic T cells [CTLs], CD8^+ Texs, and cycling CD8^+ T cells), and 2 were NK cell subclusters (*FCGR3A*^+ NK cells and *XCL2*^+ NK cells) (Fig. 5a, b). Compared with the RA groups, the IA_act groups displayed significantly higher fractions of CD8^+ Texs, CD8^+ Tems and CD4^+ Tregs but significantly lower fractions of CD4^+ Tems and CD4^+ Texs (Fig. 5c, Supplementary Fig. 10). The identity of each T/NK cell subcluster was labeled based on marker genes of the T/NK cell lineage and functions (Supplementary Fig. 11a, b). To comprehensively understand the differences in T cells between IA_act and RA, we evaluated the expression of panels of genes in 4 categories, namely, exhaustion markers (*PDCD1, CTLA4*), inflammation markers (*IFNG, CXCL13*), trafficking markers (*CXCR3, CXCR6*) and cytotoxicity markers (*GZMB, PRF1*), in subclusters of interest (Fig. 5d, e; Supplementary Fig. 10c). Generally, the expression of these markers was significantly higher in T cells in IA_act than those in RA, suggesting more functional states of T cells were present in IA_act than in RA. An exception was observed in CD8^+ Tems, which were significantly enriched in IA_act but exhibited insignificant differences in the expression of these markers between IA_act and RA.

Given that CD8^+ Texs were nearly exclusive to SFMCs of IA_act patients, we extracted the SFMC-derived CD8^+ Texs in the IA_act group and further subgrouped them into 4 clusters (C1–C4) to explore potential heterogeneity (Fig. 5f). Intriguingly, CD8^+ Tex-C1 was marked by elevated expression of terminal exhaustion markers (*PDCD1, ENTPD1, HAVCR2, TOX*), whereas CD8^+ Tex-C4 was marked by elevated

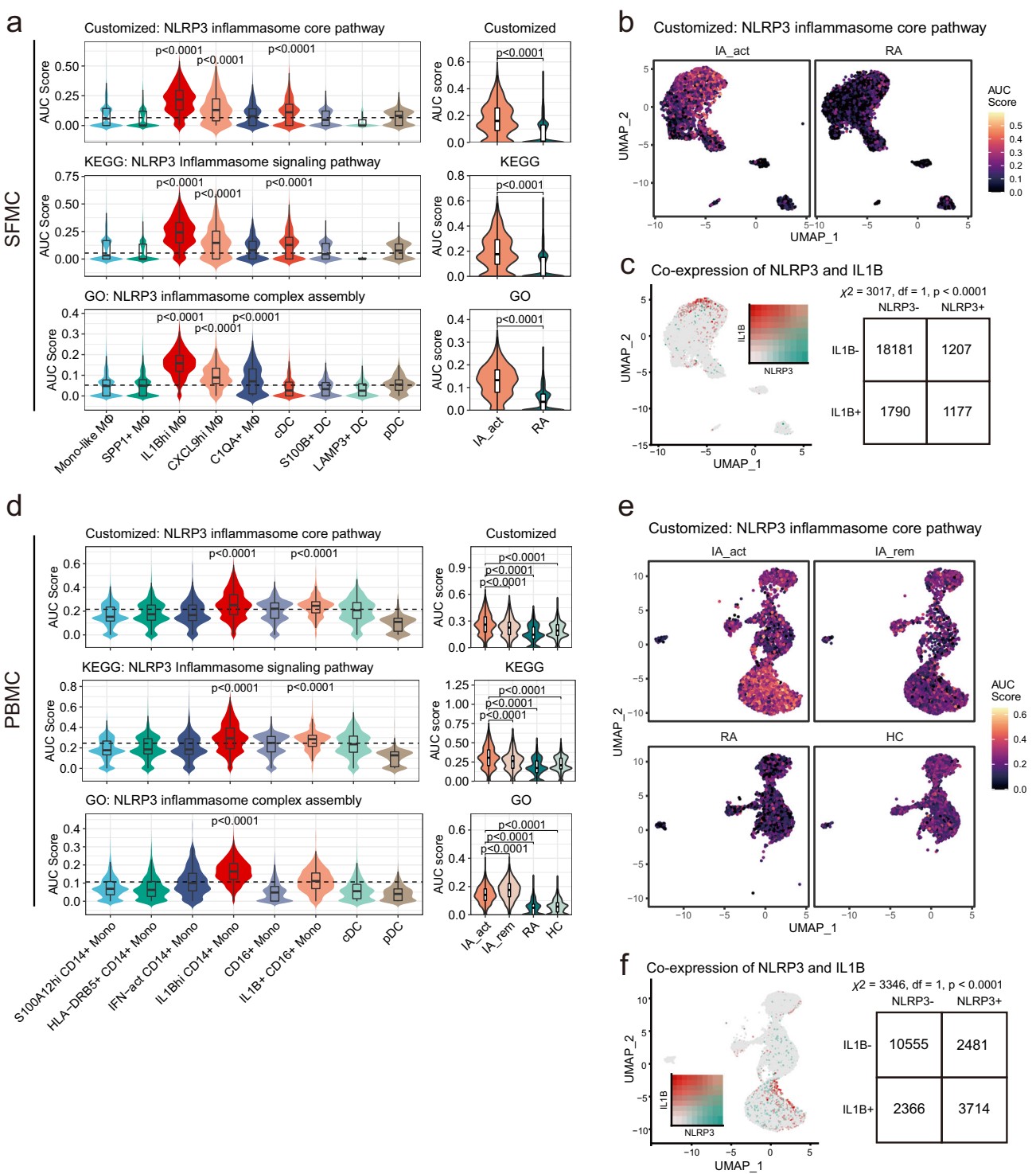

expression of progenitor exhausted cell markers (*PDCD1*, *TCF7*, *IL7R*, *SELL*) (Fig. 5g). In cancer immunotherapy models, CD8+ T cells have been reported to include a subset containing a progenitor exhausted population, which includes cells that are self-renewing, promote tumor control induced by ICIs, and eventually differentiate into a terminally exhausted population[20]. To validate whether CD8+ Tex-C4 exhibits the progenitor-exhausted phenotype, we scored each cell with well-established terminally and progenitor-exhausted gene signatures (Supplementary Table 3). The results revealed that CD8+ Tex-C4 exhibited significantly higher activity for the progenitor exhausted signature. Conversely, CD8+ Tex-C1 was significantly enriched in the terminally exhausted signature (Fig. 5h).

Although the cell fraction of cycling CD8+ T cells did not differ between the IA_act and RA groups, we identified distinct exhaustion statuses for cycling CD8+ T cells between the two groups, suggesting the existence of exhausted subpopulations related to IA_act (Fig. 5d). Therefore, we performed graph-based subclustering of cycling CD8+ T cells and obtained 4 distinct clusters (C1–C4) (Fig. 5i). Notably, CD8+ T cycling-C3 and CD8+ T cycling-C4, which were associated with IA_act (Fig. 5j), displayed a significantly elevated activity score for the terminally exhausted signature (Fig. 5k). Mirroring CD8+ Tex-C1, CD8+ T cycling-C4 was characterized by high transcriptomic expression of *CXCL13*, *TOX*, *IFNG*, *CCL3*, and *CCL4* (Fig. 5l) and exhibited a much stronger proliferative capacity indicated by the expression of MKI67

**Fig. 4 | Activated NLRP3 inflammasome signaling was characteristic of *IL1B*ʰⁱ myeloid cells and downregulated after treatment. a** The differences in NLRP3 inflammasome-related pathway signature scores among subclusters of myeloid SFMCs (left) and between IA_act and RA (right). The AUC was calculated by the AUCell algorithm (see the "Methods" section). **b** UMAP visualization of customized NLRP3 inflammasome core pathway signature scores across myeloid cells in SFMC in the IA_act and RA groups. **c** Left, UMAP visualization of the coexpression of the *IL1B* gene and *NLRP3* gene in myeloid SFMCs. Right, a two-sided chi-square test examining the association of *NLRP3*⁺ cells with *IL1B*⁺ cells in myeloid SFMCs. **d** The differences in NLRP3 inflammasome-related pathway signature scores among subclusters of myeloid PBMCs (left) and patient groups (right). The AUC was calculated by the AUCell algorithm (see the "Methods" section). **e** UMAP visualization of the customized NLRP3 inflammasome core pathway signature score across myeloid cells in PBMCs in the IA_act, IA_rem, RA, and HC groups. **f** Left, UMPA visualization of the coexpression of the *IL1B* gene and *NLRP3* gene in myeloid PBMCs. Right, a two-sided chi-square test examining the association of *NLRP3*⁺ cells with *IL1B*⁺ cells in myeloid PBMCs. Two-sided Wilcoxon tests comparing one subcluster and all other clusters were applied for the data in (**a**) and (**d**). In the box-plots of (**a**) ($n = 6$) and (**d**) ($n = 13$), the center lines denote the median of the AUC score; the lower and upper limits of the boxes denote the 25% and 75% quantile, respectively. Two-sided Wilcoxon tests comparing the median AUC score of one subcluster with all other clusters. The horizontal dashed line denotes the median AUC score of all the cell clusters, and only the clusters with significantly increased medians were denoted with *p*-value. IA_act active inflammatory arthritis, IA_rem inflammatory arthritis in remission, RA rheumatoid arthritis, HC healthy control, Mono-like monocyte-like, MΦ macrophages.

(Fig. 5m). Together, these observations indicated that self-renewing progenitor exhausted CD8⁺ Texs (CD8⁺ Tex-C4) and proliferative CD8⁺ Texs (CD8⁺ T cycling-C4) could both contribute to the pathogenesis and especially the persistence of PD-1-IA.

### *IL1B*ʰⁱ myeloid cells orchestrate cell communication through the CCR1-CCL5/CCL3 and CXCL10-CXCR3 axes

To further elucidate the crosstalk between the expanded *IL1B*ʰⁱ myeloid cell and T cell clusters in PD-1-IA, we utilized CellPhoneDB[21], a ligand–receptor interaction database, to identify cytokine–receptor and chemokine–receptor interaction pairs in both the myeloid cell and T-cell populations. In the synovial fluid, myeloid cells in the PD-1-IA group displayed much stronger interactions with T cells than did those in the RA group (Fig. 6a). Generally, in the peripheral blood, the interaction pairs of cytokines/chemokines and receptors in the PD-1-IA group outnumbered those in the RA and HC groups, and the number was slightly higher in the IA_act group than in the IA_rem group (Fig. 6b). In both the peripheral blood and synovial fluid, the *IL1B*ʰⁱ clusters exhibited strong interactions with multiple T-cell subsets including CD4⁺ and CD8⁺ Tems, CD4⁺ and CD8⁺ Texs and cycling CD8⁺ T cells. Overall, in the PD-1-IA group, *IL1B*ʰⁱ myeloid cells centered the immune response through interactions with T cells, both systemically and locally at inflamed joints.

We further investigated the cytokine/chemokine–receptor pairs employed by *IL1B*ʰⁱ myeloid cells. The top 50 interaction pairs in PD-1-IA-associated clusters were filtered and plotted (Supplementary Fig. 12a). The dominant cytokine/chemokine–receptor pairs were CCR1-CCL5/CCL3 and CXCL10-CXCR3 (Fig. 6c). We validated the expression of CCR1-CCL5/CCL3 and CXCL10-CXCR3 in the IA_act group and further compared it with that in the IA_rem, RA, ICI-nonAE, and HC groups through ELISA and flow cytometry. The median synovial fluid level of CCL5 was higher in patients with IA_act than in RA patients, although the difference was not statistically significant due to the limited sample size (Fig. 6e). Notably, the serum level of CCL5 in patients with IA_act was significantly higher than that in patients with IA_rem or that in controls (Fig. 6d). Similarly, the percentages of both CCL3⁺ CD14⁺ peripheral monocytes and CCL3⁺ CD11b⁺ synovial macrophages in IA_act exceeded those in RA (Supplementary Fig. 12b–d, gating strategy in Supplementary Fig. 13a, b). As the receptor for CCL5 and CCL3, CCR1 also had higher expression in synovial CD11b⁺ macrophages and peripheral CD14⁺ monocytes in IA_act compared with those in RA and controls (Fig. 6f–h, gating strategy in Supplementary Fig. 13a, b). The ex-vivo chemotaxis assay validated that the peripheral monocytes isolated from IA_act patients showed increased transwell migration under the CCL5 treatment (Fig. 6i). Additionally, increased migration of the monocytes towards the CD8⁺ T cells was observed in IA_act patients, and the enhanced migration was attenuated through the CCL5 blockade (Fig. 6j, k). Consistently, for the CXCL10-CXCR3 interaction pair, flow cytometry indicated that CXCL10⁺ CD11b⁺ macrophages were also expanded in the synovial fluid, and CXCR3⁺ T cells accumulated in the peripheral blood of IA_act patients (Supplementary

Fig. 12e–h, gating strategy in Supplementary Fig. 13a, c). In the chemotaxis assay, CD8⁺ T cells from IA_act showed significantly enhanced migration with CXCL10 treatment (Supplementary Fig. 12i). Overall, we demonstrated that CCR1-CCL5/CCL3 and CXCL10-CXCR3 were key signaling pairs between *IL1B*ʰⁱ myeloid cells and CD8⁺ Texs in PD-1-IA.

## Discussion

Here, we report a comprehensive single-cell analysis of immune cells in both the peripheral blood and SF of PD-1-IA patients, RA patients, and healthy individuals, identifying disease-specific cell clusters related to PD-1-IA. We found that *IL1B*ʰⁱ macrophages and CD8⁺ Texs may play key roles in the pathogenesis of PD-1-IA, providing insights into this disease (Fig. 7).

Although PD-1-IA has RA-like characteristics, such as polyarthritis, it is clinically and serologically distinct from RA. PD-1-IA patients are not predominantly female, have fewer erosive changes, and are more often seronegative[22,23]. However, a detailed immunological comparison between PD-1-IA and RA to guide clinical strategies is lacking. In this study, we compared PD-1-IA with RA and observed several abnormalities in myeloid cells in PD-1-IA, including strikingly enriched *IL1B*ʰⁱ macrophages in SFMCs and *IL1B*ʰⁱ CD14⁺ monocytes in PBMCs. IL1β, as an important inflammatory mediator of the innate immune response, participates in the pathogenesis of chronic inflammatory diseases[24]. In RA, IL1β levels are locally elevated[25]. Our results demonstrated that *IL1B*ʰⁱ myeloid cells were much more expanded in PD-1-IA than in RA, implying a severe inflammatory reaction. IL1β is thus a potential target for the treatment of PD-1-IA, particularly because IL1β blockade synergizes with anti-PD-1 therapy for tumor abrogation[26,27].

Trajectory analysis indicated that *IL1B*ʰⁱ macrophages likely differentiated from *IL1B*ʰⁱ CD14+ monocytes in PBMCs. This hypothesis was supported by the elevated expression of chemokine receptor genes, especially *CCR1*, on synovial macrophages and peripheral CD14⁺ monocytes in PD-1-IA, suggesting the recruitment of peripheral monocytes to the synovial fluid. Moreover, CCL5, the ligand of CCR1, was mainly produced by CD8⁺ CTLs and found to be enriched in the serum of PD-1-IA patients compared with controls. Therefore, we hypothesized that PD-1 inhibitors activate CD8⁺ T cells, promoting the secretion of CCL5, which recruits peripheral *CCR1*⁺ *IL1B*ʰⁱ CD14⁺ monocytes to the synovial fluid. Then, *CCR1*⁺ *IL1B*ʰⁱ CD14⁺ monocytes differentiate into *CCR1*⁺ *IL1B*ʰⁱ macrophages and initiate an inflammatory response. CCR1 is also a potential target for alleviating joint inflammation in PD-1-IA.

Compared with IA_act, IA_rem showed reduced *IL1B* gene expression in CD14⁺ monocytes, which was validated by flow cytometry, although the statistical significance was borderline ($p = 0.0571$) due to the limited sample size. Therefore, peripheral *IL1B*ʰⁱ CD14⁺ monocytes are associated with disease activity and may be a useful biomarker for disease monitoring. We also observed that the aberrantly activated TNF signaling pathway was downregulated in peripheral *IL1B*ʰⁱ CD14⁺ monocytes of PD-1-IA patients after standard pan-anti-inflammatory treatment, while other activated proinflammatory

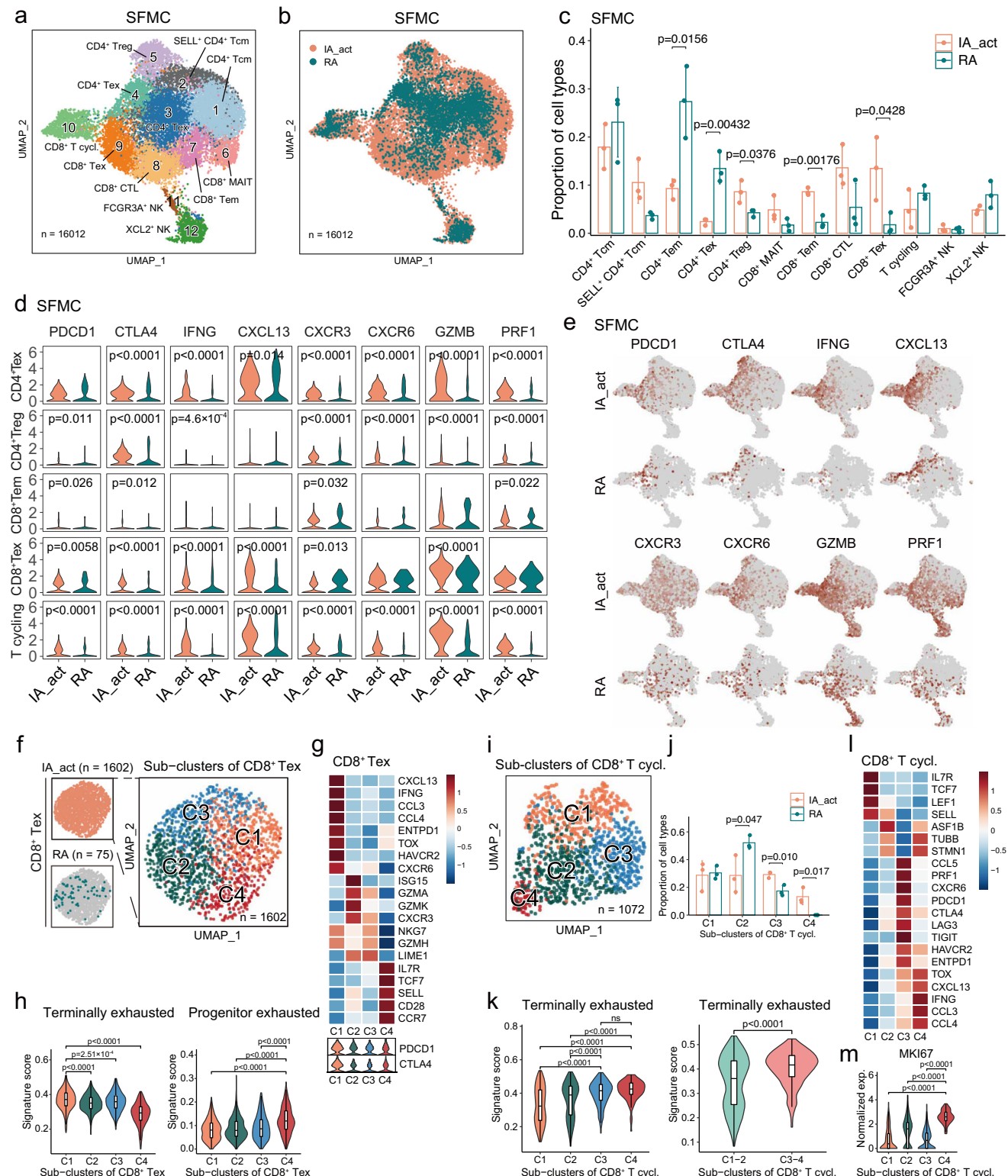

pathways were not significantly altered, indicating that targeting TNF is a more precise approach than pan-anti-inflammatory treatment. Given that TNF has recently been shown to be a promising target for preventing irAEs without impeding ICI efficacy in mouse colitis and cardiotoxicity models[28,29], targeting TNF is likely to uncouple toxicity and anti-PD-1 efficacy in PD-1-IA.

Activated T cells have been extensively reported to be present in tissues affected by other irAEs, such as colitis[30], thyroiditis[31], dermatitis[32], and hepatitis[33], which may explain the rapid response to

glucocorticoids, as glucocorticoids induce apoptosis in activated T cells[34]. A recent study reported a prominent IFNγ-producing CD8+ T-cell axis in both the blood and synovial fluid in ICI-induced arthritis[35], suggesting a similar immunophenotype in ICI-induced arthritis, although it is difficult to explain the long course of this arthritic condition. Our study revealed that the progenitor-exhausted CD8+ T-cell population in the synovial fluid was a disease-specific cluster related to PD-1-IA. Considering that progenitor-exhausted CD8+ T cells persist long-term and respond to anti-PD-1 treatment in melanoma mouse

**Fig. 5 | Heterogenous patterns of CD8⁺ T-cell exhaustion in the SFMCs of PD-1-IA patients. a** Identification of 12 subclusters of T and NK cells across all SFMC samples. **b** Distribution of T/NK cell subclusters of SFMCs between IA_act and RA. **c** Quantification of the fraction (mean ± SD) of each T/NK subcluster between the patient groups with a two-sided *t*-test. **d** Violin plots showing the expression of genes of interest in selected T/NK cell subclusters between IA_act and RA with a two-sided Wilcoxon test. **e** UMAP plots showing the expression of genes of interest in T/NK cell subclusters between IA_act and RA. **f** Subclustering of 1602 exhausted CD8⁺ T cells from the SFMCs of IA_act patients into 4 subsets. **g** Heatmap showing the expression of marker genes across subsets of exhausted CD8⁺ T cells (CD8⁺ Texs) (top) and violin plot showing the expression of the *PDCD1* and *CTLA4* genes across subsets of CD8⁺ Texs (bottom). **h** Quantification of terminally exhausted (left) and progenitor exhausted signature scores (right) among CD8⁺ Texs with an unpaired two-sided Wilcoxon test. Signature scores were calculated by the AUCell

algorithm (see the "Methods" section). **i** UMAP plot showing 1072 cycling CD8⁺ T cells in SFMCs from all individuals separated into 4 subsets. **j** Quantification of the subcluster proportions (mean ± SD) between IA_act and RA with a two-sided *t*-test. **k** Quantification of the differences in the terminally exhausted signature score among 4 subsets of cycling CD8⁺ T cells (left) and between the C1-2 subset and C3-4 subset (right) with a two-sided Wilcoxon test. **l** Heatmap showing the expression of marker genes across subsets of cycling CD8⁺ T cells. **m.** Quantification of the *MKI67* gene expression between the C4 subset and other subsets with a two-sided Wilcoxon test. In the box-plots of (**h**), (**k**), and (**m**), the center lines denote the median of the signature score or normalized gene expression in each subset; the lower and upper limits of the boxes denote the 25% and 75% quantile, respectively. IA_act active inflammatory arthritis, IA_rem inflammatory arthritis in remission, RA rheumatoid arthritis, HC healthy control, T.cycl. cycling T cells.

models[36], these cells in the synovial fluid may also persist long-term and contribute to the long course of PD-1-IA.

This study had several limitations. First, the number of patients from whom samples were collected for scRNA-seq was relatively small due to the rarity of PD-1-IA, which led to the lack of statistical power. Second, only two synovial fluid samples were used for validation in this study, which prevented meaningful statistical analysis. This is because it is not common for a PD-1-IA patient to simultaneously meet the requirements of having a large amount of joint effusion, meeting the indication for joint puncture, and giving informed consent. These issues will be addressed in our future studies by collecting more eligible individuals.

In summary, our study provides a comprehensive immunologic landscape of PD-1-IA patients and identifies several cellular and molecular abnormalities related to PD-1-IA. Our findings may shed new light on the pathogenesis of PD-1-IA, providing clues for biomarkers of disease activity and therapeutic targets that preserve antitumor efficacy.

## Methods
### Study design and participants
We prospectively recruited adult patients with new-onset IA induced by PD-1 inhibitors (PD-1-IA). Active PD-1-IA (IA_act) was defined as[4]: (1) the presence of joint inflammation diagnosed by a rheumatologist based on the comprehensive assessment of the history, physical examination, inflammatory markers, and imaging findings; (2) joint inflammation developed after anti-PD-1 administration. IA patients were excluded if they had a pre-existing autoimmune disease. We defined remission of IA (IA_rem) as the absence of clinical features of active joint inflammation after treatment of IA. Patients who fulfilled the 2010 American College of Rheumatology/European League Against Rheumatism (ACR/EULAR) classification criteria for RA were recruited and further classified into seropositive or seronegative RA depending on the presence or absence of anticitrullinated-peptide antibodies (ACPA) and rheumatoid factor (RF). Additionally, healthy donors with matched age and sex were recruited as controls. All patients and healthy donors provided informed consent for the collection of research samples. Our study was approved by the Peking Union Medical College Hospital Ethics Committee (no. JS-1940).

Fresh synovial fluid samples were obtained from patients with active IA or active RA through arthrocentesis. Prior to arthrocentesis, paired peripheral blood samples were collected from the patients with active IA or active RA. In addition, peripheral blood was collected from patients with IA_rem, ICI-treated patients without irAEs (ICI-nonAE), and healthy donors. Single-cell data of the synovial fluid from RA patients and the peripheral blood from RA patients and healthy donors in this study were retrieved from the previously published data in our research group (assession number "HRA000155")[37]. The single-cell dataset of the PBMCs from the ICI-nonAE was retrieved from a previously published article on irAEs (assession number "GSE180045")[11].

### Sample preparation and cell sorting
Synovial fluid mononuclear cells (SFMCs) and peripheral blood mononuclear cells (PBMCs) were isolated from synovial fluid and peripheral blood, respectively, using Ficoll-Paque gradient centrifugation. CD45⁺ mononuclear cells were isolated using magnetic beads. Cell quantity and viability were assessed using Trypan Blue staining.

### Library preparation and scRNA-seq
Single-cell 3' gene expression libraries for CD45⁺ immune cells were prepared by strictly following the protocols of a Chromium Single Cell 3'v2 Library kit (10x Genomics). All resulting libraries were sequenced on the Illumina NovaSeq6000 platform (Novogene and Berry Genomics, Beijing, China).

### Data alignment and quality control
Raw data were assembled using Illumina's bcl2fastq converter. After initial quality control, adaptor sequences, sequences with over three uncertain nucleotides (designated as N) and low-quality reads were removed. Only clean data for high-quality reads were aligned to the human reference genome with the 10x Genomics Cell Ranger pipeline using default parameters. Unique molecule identifiers (UMIs) for each gene and each cell barcode were counted and added to gene expression matrices. Doublets detected by Scrublet. Doublet clusters expressing markers of more than two major cell lineages, and cells meeting the following criteria were removed: (1) number of genes detected per cell <200 or >5000; (2) UMIs detected per cell over 200; (3) counts of mitochondrial genes constituting over 12% of all gene counts; and (4) counts of red blood cell genes constituting over 0.3% of all gene counts. Finally, we obtained 40,482 SFMCs and 64,819 PBMCs for downstream analyses.

### Dimensionality reduction and annotation of major cell clusters
The gene expression matrices of each sample were analyzed and initially clustered for major cell subsets. Briefly, normalization and dimensionality reduction were performed on integrated matrices of SFMCs and PBMCs using the Seurat package. The batch effects in the study samples were removed by the *RunHarmony* function in the Harmony package. The uniform manifold approximation and projection (UMAP) method was applied to visualize a 2D projection of cell clusters from an SNN graph. Cells were clustered with highly variable genes at 0.7 resolution. The top 30 differentially expressed genes (DEGs) of each cluster were identified and carefully examined for well-recognized marker genes. Based on the marker genes of each cluster, two researchers manually annotated the cell clusters and rechecked them independently.

### Subclustering of myeloid cells and T/NK cells
To investigate myeloid cells and T cells in detail, we performed subclustering analysis on the SFMC and PBMC datasets.

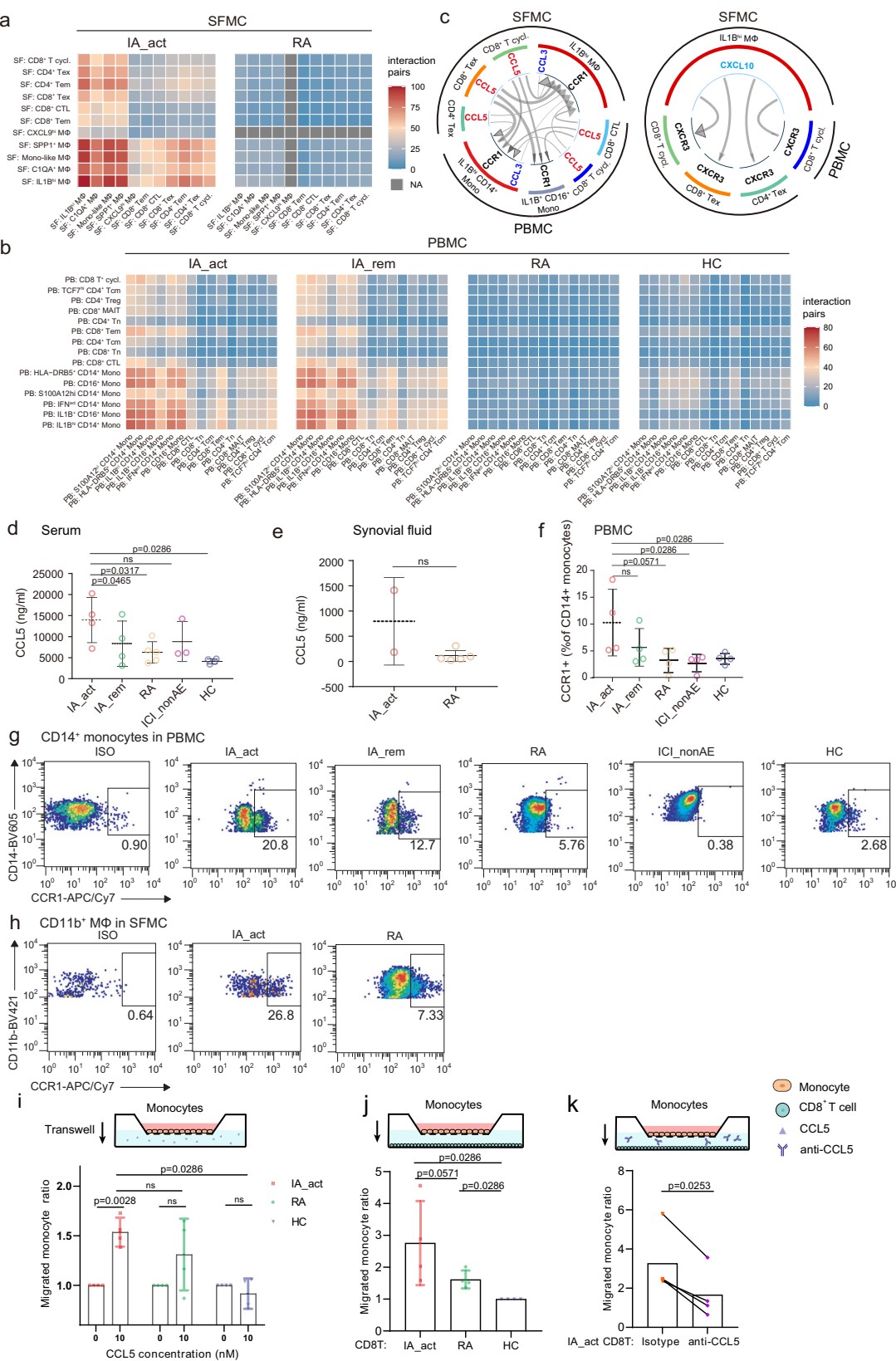

We extracted the myeloid cell and T-cell clusters of the original dataset and then reclustered and annotated them following the pipelines described above. The proportions of cell clusters in the dataset were compared across sample sets using a paired or unpaired two-tailed Student's t-test, as stated. For cycling T cells and exhausted CD8+ T-cell subsets in SFMCs, we performed the same subclustering procedures. To reveal the enrichment of exhausted features in these cell clusters, signature scores were calculated at the single-cell level based on the expression of customized exhausted gene sets using the AUCell package[38]. The differences between cell clusters were assessed by a two-tailed Student's t-test. p < 0.05 was considered statistically significant.

**Fig. 6 | Crosstalk between CD8+ Texs and *IL1B*hi myeloid cells. a** and **b** Number of interaction pairs between monocytes and T cells in PBMCs (**a**) and in SFMCs (**b**) in each patient group. All the interactions were statistically significant by permutation tests. **c** Significant chemokine ligand–receptor pairs across *IL1B*hi myeloid cells and T cells in PBMCs and SFMCs. CCL5-CCR1 and CCL3-CCR1 (left) and CXCL10-CXCR3 (right) ligand–receptor pairs are presented separately. The direction of the arrows indicates the interaction between the ligand to the receptor. **d** and **e** Quantification of the serum (**d**) and synovial fluid CCL5 (**e**) protein concentration (mean ± SD) among patient groups. The data show $n = 2–5$ biological replicates over two independent experiments. **f** Flow cytometry showing the percentages (mean ± SD) of CCR1+ CD14+ monocytes in PBMCs among patient groups ($n = 4$ per group, gating strategy in Supplementary Fig. 13b). The data show $n = 4$ biological replicates over three independent experiments. Paired two-sided *t*-test compared the IA_act with IA_rem in (**d**) and (**f**). Two-sided Wilcoxon tests compared the IA_act with the RA, ICI-nonAE, or HC in (**d**–**f**). **g** Representative flow cytometry plots for (**f**). **h** Flow

cytometry showing the percentages of CCR1+ macrophages in SFMCs in IA_act and RA patients (gating strategy in Supplementary Fig. 13a). **i** Transwell migration of monocytes under the CCL5 treatment by the chemotaxis assay. The migrated cell ratio was the division of migrated cell counts with CCL5 treatment to those without CCL5 treatment, comparing with ratio paired-*t* tests within the patient groups, and unpaired two-sided Wilcoxon tests among the patient groups (IA_act, HC, and RA). **j** Transwell migration of monocytes towards T cells among the patient groups (IA_act, HC, and RA), comparing with unpaired two-sided Wilcoxon tests. **k** Transwell migration of monocytes towards T cells in IA_act, with or without CCL5 blockade, comparing with a two-sided ratio paired *t*-test. The data show $n = 4$ biological replicates over three independent experiments. ns nonsignificant, Mono-like monocyte-like, IA_act active inflammatory arthritis, IA_rem inflammatory arthritis in remission, RA rheumatoid arthritis, HC healthy control, MΦ macrophages, Mono monocytes, T.cycl. cycling T cells.

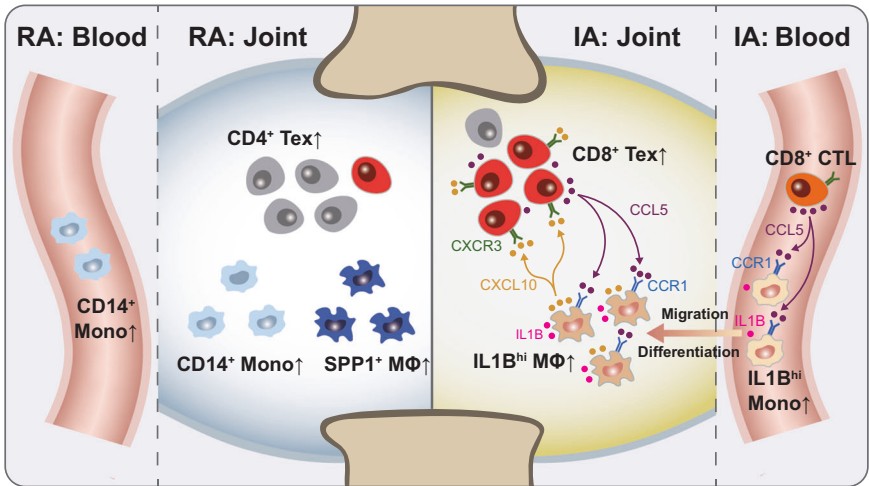

**Fig. 7 | Graphical summary of this study.** In IA, synovial *IL1B*hi macrophages, likely originating from peripheral *IL1B*hi CD14+ monocytes, communicate with exhausted CD8+ T cells through the CCR1-CCL5/CCL3 and CXCL10-CXCR3 axes (right). This

interaction may play a key role in PD-1-IA pathogenesis but not in RA (left). RA rheumatoid arthritis, IA inflammatory arthritis, MΦ macrophage, Mono monocyte, Tex exausted T cell, CTL cytotoxic T cell.

## DEG analysis

To elucidate the differences in gene expression profiles among cell clusters and between disease and control samples, DEGs were analyzed. DEGs were identified using *FindAllMarkers* functions of the Seurat package with the Wilcoxon test and adjusted *p* values with the Bonferroni correction. DEGs were filtered with a maximum adjusted *p*-value of 0.05 and a minimum percentage of cell expression of 0.1 and ranked by the average log-transformed fold change. Only the top 10 and bottom 10 ranked DEGs are shown in the graph.

## Functional enrichment analysis

Functional enrichment analysis of DEGs across clusters and sample sets was performed using the *GSEA* function of the ClusterProfiler package[39]. Human hallmark gene sets downloaded from the Molecular Signatures Database[15] (MSigDB, https://www.gsea-msigdb.org/gsea/msigdb/human/collections.jsp) were used as reference gene sets. Functionally enriched pathways were defined as processes with a maximum adjusted *p*-value of 0.05. Signature scores (AUCscore) for the enriched pathways were calculated at the single-cell level based on the expression of the MSigDB gene sets using the AUCell package[38].

## Trajectory analysis

To identify cell transitions, single-cell trajectory analysis was performed using the *Monocle3* package. We combined the monocyte

subsets and T/NK cell subsets of the SFMC and PBMC datasets, respectively. Batch effects among samples were removed using the *align_cds* function of the Monocle3 package, which applied the mutual nearest neighbor alignment technique by calling the package batchelor[40]. Then, single-cell trajectories were constructed by pseudotime analysis. Along the *IL1B*hi monocyte/macrophage trajectory, gene expression profiles were tracked to determine highly variable genes. The top 50 DEGs along the IL1Bhi trajectory (arranged by pseudotime) were identified and organized into functional modules by the *find_gene_modules* function of the Monocle3 package.

## Analysis of cell–cell interactions based on ligand–receptor expression

Interactions between cell clusters were examined based on ligand–receptor expression profiles. We applied the CellPhoneDB database (version 3.0.1)[21] for the initial assessment of cell–cell interactions. The normalized ligand–receptor expression of each sample was individually used as the input for CellPhoneDB analysis, and the results were integrated according to the sample sets. We focused on cytokine and chemokine receptors. We applied LRPlot of the iTalk package to visualize the expression of specific cytokine/chemokine–receptor pairs across the IA-associated cell clusters[41]. In LRPlot, the width of the arrow (arrowhead) represents the expression level of the ligand (receptor).

## Flow cytometry

SFMCs from IA_act or RA patients and PBMCs from IA_act, IA_rem, RA, and ICI-nonAE patients and HCs were collected. Cells were washed twice with FACS staining buffer (PBS, 5% FBS, and 0.1% sodium azide) and stained with antibodies for 30 min on ice. The following antibodies were used: anti-CD14-BV605 (#367126, clone 63D3, 1/20 dilution, BioLegend), anti-CD11b-BV421 (#301323, clone ICRF44, 1/20 dilution, BioLegend), anti-CD4-APC/Cy7 (#344615, clone SK3, BioLegend), anti-CD8-BV510 (#344731, clone SK1, 1/20 dilution, BioLegend), anti-IL-1β-FITC (#508206, clone JK1B-1, 1/20 dilution, BioLegend), anti-CCL3-PE (#12-9706-42, clone CR3M, 1/20 dilution, Thermo Fisher), anti-CCR1-APC/Cy7 (#362917, clone 5F10B29, 1/20 dilution, BioLegend), anti-CXCL10-APC (#519505, clone J034D6, 1/20 dilution, BioLegend), and anti-CXCR3-APC (#353707, clone G025H7, 1/20 dilution, BioLegend). For the detection of intracellular IL-1B, CCL3, CXCL10, and CXCR3 expression, cells were first stimulated with phorbol myristate acetate and ionomycin for 4 h with GolgiStop in complete RPMI-1640 medium in an incubator at $37\,°C$ with 5% $CO_2$. The stimulated cells were then fixed and permeabilized with a fixation/permeabilization kit (eBioscience) before intracellular staining. Stained cells were analyzed on a FACSCelesta™ (BD Biosciences). The gating strategies for monocytes and macrophages are shown in Supplementary Fig. 13. Data analysis was carried out using FlowJo™ v10.8 Software (BD Life Sciences). Differences between groups were assessed by a paired or unpaired Student's t-test. $P < 0.05$ was considered statistically significant.

## Enzyme-linked immunosorbent assay (ELISA)

For ELISA tests, synovial fluid and peripheral blood samples were collected using coagulation-promoting tubes, and the supernatant was collected after centrifugation at $1800 \times g$ for 10 min. The CCL5 levels in synovial fluid and serum were measured with the Human CCL5 ELISA Kit (Abcam ab174446) following the manufacturer's protocol. Serum was diluted 1:1000 with assay buffer, and synovial fluid was diluted 1:5 for CCL5 detection. Absorbance was measured using a microplate reader (Biotek SynergyNeo2), and the final concentration was determined from standard curves. Differences between groups were assessed by a paired or unpaired Student's t-test. $p < 0.05$ was considered statistically significant.

## Chemotaxis assay

For chemotaxis assay, monocytes and CD8[+] T cells were isolated from PBMCs of IA, RA, and HCs with magnetic beads (CD14 or CD8 microbeads, Miltenyi Biotec) and resuspended at $5 \times 10^6$ cells/mL in RPMI 1640 containing 0.5% bovine serum albumin (BSA). For the chemokine-cell chemotaxis assay, cells (monocytes or CD8[+] T cells) were seeded in the upper chambers of the 24-well Transwell chambers with 5 μm pores (Corning, NY), and the chemokine (CCL5 or CXCL10) was placed in the lower wells at 0 or 10 ng/ml, and incubated at $37\,°C$ for 2 or 24 h. For the cell-cell chemotaxis assay, monocytes prestained with anti-CD14-APC (#301807, clone M5E2, 1/20 dilution, BioLegend) were seeded in the upper chambers and CD8[+] T cells prestained with anti-CD8-PE-Cy7 (#344712, clone SK1, 1/20 dilution, BioLegend) were seeded in the lower wells. To investigate the role of CCL5 in the T-monocyte chemotaxis, CCL5 neutralizing antibody (MAB278, R&D) or its isotype (Mouse IgG1) were added to the lower wells. After incubation for 24 h at $37\,°C$, migrated cells were collected from the bottom wells, washed once with PBS, and counted by fluorescence-activated cell sorting technique (Attune NxT). The gating strategies for migrated cells are shown in Supplementary Fig. 14. The migration rate was calculated as the ratio of migrated cells to the total seeded cells. The migration rates were compared among the groups using an unpaired Wilcoxon test or ratio paired-t test.

## Statistics and data visualization

Continuous variables with non-normalized distribution are expressed as the median and interquartile range (IQR). Statistical analysis and graphing were performed with R software (version 4.1.2, R Core Team), R package ggplot2 (version 3.3.5), GraphPad Prism (version 8.0.1, GraphPad Software, CA, USA), and Adobe Illustrator 2023 (Adobe). As indicated, data were analyzed using a two-sided Wilcoxon test or paired/unpaired two-sided t-test for double-group comparison, and one-way ANOVA for multi-group comparisons. As stated in the figure legends, p-values were adjusted for multiple comparisons using Benjamini–Hochberg (BH) correction.

## Reporting summary

Further information on research design is available in the Nature Portfolio Reporting Summary linked to this article.

## Data availability

The RNA sequencing datasets generated in this study have been deposited in the Genome Sequence Archive (GSA) under the accession code HRA004500. Source data are provided with this paper.

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

## Acknowledgements

This study was supported by grants from the National Key R&D Program of China (2022YFC3602000, X.Z.), the Chinese Academy of Medical Science Innovation Fund for Medical Sciences (CIFMS) (2023-I2M-C&T-B-041, H.Y., 2021-I2M-1-016, H.Y., 2021-I2M-1-017, X.Z., 2020-I2M-C&T-B-011, H.Y., 2021-I2M-1-047, M.W.), National High-Level Hospital Clinical Research Funding (2022-PUMCH-A-106, H.Y., BJ-2022-116, X.Z., BJ-2023-084, M.W.), Chinese Society of Clinical Oncology (CSCO)-Pilot Oncology Research Fund (Y-2019AZMS-0452, H.Y.), National Natural Science Foundation of China (82230060, X.Z., 81788101, X.Z., 32141004 X.Z, 82203021 X.Z., 82171799, X.J.), and Peking Union Medical College Hospital Research funding for Postdoc (kyfyjj202317, Z.Z.).

## Author contributions

H.Y. and X.Z. designed and supervised the study; Z.Z., B.Y., X.L., L.Z., Y.F., H.C., L.Z., X.S., N.L. and Y.W. collected samples and performed the clinical analysis; Z.Z., X.Z., X.J., D.Y., Y.P. and Y.Y. designed and performed experiments; Z.Z., X.Z., X.J., Z.Y., Y.H., X.W. and M.W. performed bioinformatic analysis; Z.Z., X.Z., X.J., H.Y. and X.Z. drafted and revised the manuscript.

## Competing interests

The authors declare no competing interests.

## Additional information

[1]Department of Rheumatology and Clinical Immunology, National Clinical Research Center for Dermatologic and Immunologic Diseases, the Ministry of Education Key Laboratory, Peking Union Medical College Hospital, Chinese Academy of Medical Sciences and Peking Union Medical College, 100730 Beijing, China. [2]Clinical Immunology Center, Chinese Academy of Medical Sciences and Peking Union Medical College, 100730 Beijing, China. [3]State Key Laboratory of Complex Severe and Rare Diseases, Peking Union Medical College Hospital, Chinese Academy of Medical Sciences and Peking Union Medical College, 100730 Beijing, China. [4]Department of Thoracic Surgery, National Cancer Center/National Clinical Research Center for Cancer/Cancer Hospital, Chinese Academy of Medical Sciences and Peking Union Medical College, 100021 Beijing, China. [5]National Infrastructure for Translational Medicine, Institute of Clinical Medicine, Peking Union Medical College Hospital, Chinese Academy of Medical Sciences and Peking Union Medical College, 100730 Beijing, China. [6]Department of Orthopedics, Peking Union Medical College Hospital, Chinese Academy of Medical Sciences and Peking Union Medical College, 100730 Beijing, China. [7]Department of Pulmonary and Critical Care Medicine, Peking Union Medical College Hospital, Chinese Academy of Medical Sciences and Peking Union Medical College, 100730 Beijing, China. [8]Department of Thoracic Surgery, Peking Union Medical College Hospital, Chinese Academy of Medical Sciences, 100730 Beijing, China. [9]Department of Thoracic Oncology, Cancer Center, and Laboratory of Clinical Cell Therapy, West China Hospital, Sichuan University, 610041 Chengdu, China. [10]Department of Rheumatology, Beijing Hospital, National Center of Gerontology, Institute of Geriatric Medicine, Clinical Immunology Center, Chinese Academy of Medical Sciences, 100730 Beijing, China. [11]These authors contributed equally: Ziyue Zhou, Xiaoxiang Zhou, Xu Jiang, Bo Yang. ✉e-mail: yanghuaxia2013@163.com; zhangx@bjhmoh.cn

