## [Peer Review File · Nature Communications]

REVIEWER COMMENTS

Reviewer #1 (expert in immune checkpoint inhibitor-induced arthritis):

This is a study of CD45+ cells in synovial fluid and blood of patients with inflammatory arthritis due to immune checkpoint inhibitor therapy. The authors used single cell RNAseq to analyze cell populations in these patients. Five patients with ICI-IA and 8 with RA are evaluated. The distribution of cell types was quite different between the ICI-IA and RA groups. The study focuses on a relevant and increasingly important area of rheumatic irAEs. It is very light on correlative clinical data for the included patients. There is also some concern about whether all the appropriate controls were included.

Specific comments are detailed below.

-In the introduction it is not entirely accurate to say that rheum irAEs usually persist. This has only been studied in inflammatory arthritis and in the study cited slightly less than half persisted. Would change this language.

-In the methods the authors should say more about how they defined ICI-IA.

-There is a lack of information on the clinical features of ICI-IA and how homogeneous the group studied was. This would be useful as phenotypically there are differences within ICI-IA.

-It is interesting that the authors picked only seropositive RA as a comparison. Given that patients are seronegative with ICI-IA primarily, would have been interesting to compare to seronegative RA or spondyloarthritis.

-To fully understand the role of CD8+Tex cells (or any of the cell types for that matter), it would be useful to have PBMCs from other ICI treated patients who did NOT develop ICI-IA. Without that control it is hard to say whether they are relevant specifically to ICI-IA or just to ICI treatment.

Reviewer #2 (expert in single-cell RNA sequencing):

This work describes immune cell subsets in synovial fluid (SF) and peripheral blood from anti-PD-1-induced inflammatory arthritis (IA) and rheumatoid arthritis (RA) patients using single-cell RNA-seq. Among myeloid cells in SF, SPP1+ macrophages are significantly reduced in IA compared to RA, whereas IL1B+ macrophages are highly enriched in IA. The authors also found IL1B+ classical monocytes in the blood of IA patients. Using trajectory analysis, they speculated that IL1B+ macrophages in SF are derived from IL1B+ monocytes. Both IL1B+ macrophages in SF, and IL1B+ monocytes in the blood highly express NLRP3 inflammasome signaling-associated genes. Then, the authors moved to T-cell characterization in SF of IA and RA patients, and found significant several CD8 T cell accumulations in SF of IA patients, including effector memory, exhausted and progenitor of exhausted T cells. Finally, authors performed receptor-ligand interaction analysis using single-cell data, and speculated that interaction of CCR1 and CCL5/3 and/or CXCL10 and CXCR3 may be important for IL1B+ macrophage-Tcell interaction in IA patients. The study is relatively descriptive, but it provides a valuable resource to the immunology communities. There are some concerns.

1. The major concern is that conclusions from receptor-ligand analysis are too speculative. The author should perform some functional analysis. Does CCL5 from activated CD8 T cells (anti-PD1 treated?) really induce chemotaxis of IL1B+ monocytes or macrophages? Does CXCL10 from IL1B+ monocytes or macrophages attract CD8+ T cells? Do IL1B+ macrophages induce T cell proliferation or activation?

2. To confirm T cell-macrophage interactions in SF of IA patients, imaging analysis may be helpful. Some subsets of CD8+ T cells may be heavily interacting with macrophages in SF of IA, but not in RA. Visualization of IL1B+ macrophages in SF using in situ hybridization is also interesting although it may be difficult.

3. In figure 1, the authors showed enrichment of T cells and reduction of myeloid cells in SF of IA_act. This is just an analysis of cell frequency. The authors should show the count of T cells and myeloid cells (possibly by flow cytometry?). If T cells are accumulated in SF of IA, the percentage of myeloid cells may go down. At the same time, the author can also validate the finding of figure 5 by flow cytometry (accumulation of some CD8+ T cell subsets)

4. The authors used PMA and ionomycin to detect IL-1B, and CCL3, and CXCL10 in monocytes/macrophages. PMA and ionomycin are normally used to activate T cells. Have you

detected these cytokines and chemokines in macrophages without these stimulations?

5. In figure 2, the authors showed IL1B was high in IL1B+ PB monocytes in IA-act but not those in IA_rem. What are the differences and similarities between them? DEG analysis of the subset between IA_act and IA_rem may be interesting.

6. In figure 1a, do you need to mention about ELISA? If I understand correctly, the authors used ELISA just for validation of some cytokines and chemokines and those results are not included in figure 1.

RESPONSE TO REVIEWERS' COMMENTS

Reviewer #1 (expert in immune checkpoint inhibitor-induced arthritis):

1. In the introduction it is not entirely accurate to say that rheum irAEs usually persist. This has only been studied in inflammatory arthritis and in the study cited slightly less than half persisted. Would change this language.

Reply:

Thanks for the suggestion. It is true that in the study we just mentioned IA which persisted in approximately half of patients after ICI cessation. We follow the reviewer's advice and have deleted the sentence "*Unlike other irAEs, rheumatic irAEs usually follow a chronic course, requiring long-term medication and seriously reducing the quality of life*". (page 3, line 72).

2. In the methods the authors should say more about how they defined ICI-IA.

Reply:

We have added the definition of ICI-IA (IA_act and IA_rem) in the methods section according to the reviewer's suggestion, written as the following: "We prospectively recruited new-onset adult patients with IA induced by PD-1 inhibitors (PD-1-IA). Active IA(IA_act) was defined as ¹: 1) presence of active joint inflammation diagnosed by a rheumatologist based on the comprehensive assessment of the history, physical examination, inflammatory markers and imaging findings; 2) joint inflammation developed after anti-PD-1 administration. IA patients were excluded if they had pre-existing autoimmune diseases. We defined remission of IA (IA_rem) as the absence of clinical features of active joint inflammation after treatment of IA." (page 12, line 405-411, highlighted in yellow).

3. There is a lack of information on the clinical features of ICI-IA and how homogeneous the group studied was. This would be useful as phenotypically there are differences within ICI-IA.

Reply:

We appreciate the valuable comment. We have listed the demographics, clinical characteristics, and treatment of ICI-IA in the **Supplementary Table 1-2**. According to the suggestion, we also added a detailed description of the clinical features of ICI-IA in the section of Results (page 4, line 112-115, highlighted in yellow).

Supplementary Table 1. Demographics of enrolled patients with inflammatory arthritis and rheumatoid arthritis.

Patient	Age	Gender	Current Status (Until Nov. 2022)	ACPA	RF
IA_1	30	Female	alive	Negative	Negative
IA_2	59	Female	alive	Negative	Negative
IA_3	51	Male	alive	Negative	Negative
IA_4	65	Female	alive	Negative	Negative
IA_5	59	Female	alive	Negative	Negative
RA1	64	Female	alive	Positive	Positive
RA2	32	Female	alive	Positive	Positive
RA3	62	Female	alive	Positive	Positive
RA4	53	Female	alive	Positive	Positive
RA5	58	Male	alive	Positive	Positive
RA6	73	Female	alive	Positive	Positive
RA7	47	Female	alive	Positive	Negative
RA8	39	Female	alive	Positive	Positive

Abbreviations: ICI = immune checkpoint inhibitor; ACPA = Anticitrullinated-peptide antibodies; RF = rheumatoid factor; IA = inflammatory arthritis; RA = rheumatoid arthritis.

Supplementary Table 2. Baseline clinical characteristics of enrolled patients with inflammatory arthritis.

Patient	Malignancy	ICIs	IA onset [†] (days)	Treatment cycles [‡]	CTCAE grade [‡]	Involved joints	Laboratory findings [‡]				Treatment of IA	Remission
							WBC [‡] ($\times 10^9/L$)	LY [‡] ($\times 10^9/L$)	ESR [‡] (mm/h)	hsCRP [‡] (mg/dL)		
IA_1	Esophageal squamous cancer	Pembrolizumab	30	4	3	Wrists, MCPs, PIPs, knees	7.97	1.65	96	21.12	TNF inhibitor, local GCs, NSAIDs	Yes
IA_2	Gallbladder mucinous adenocarcinoma	Camrelizumab	70	3	2	Knees	4	0.9	48	17.66	Local GCs, NSAIDs, MTX	Yes
IA_3	Esophageal squamous cancer	Camrelizumab	97	2	2	Knees	7.09	1.35	53	9.21	Local GCs, NSAIDs, MTX	Yes
IA_4	Small cell lung cancer	Pembrolizumab	387	29	3	MCPs, PIPs, knees	6.93	1.25	10	5.37	Local GCs, NSAIDs, MTX, TII	Yes
IA_5	Small cell cervical cancer	Camrelizumab	151	9	2	MCPs, PIPs, hips, shoulders	6.6	1.56	85	11.43	Local GCs, NSAIDs, MTX	Yes

^{*} IA onset was defined as the time interval between the first dose of ICI to IA onset.

[†] Treatment Cycles were defined as the number of ICI cycles administrated by IA onset.

[‡] CTCAE grade according to Common terminology criteria for adverse events.

Abbreviations: IA = inflammatory arthritis; ICI= immune checkpoint inhibitor; CTCAE = Common terminology criteria for adverse events; MCP = metacarpophalangeal joint; PIP =proximal interphalangeal joint; WBC = white blood cells; LY = lymphocyte; ESR = erythrocyte sedimentation rate; hsCRP = high-sensitivity C-reactive protein; TNF = tumor necrosis factor; NSAIDs = non-steroidal anti-inflammatory drugs; GC = glucocorticoid; MTX = methotrexate; TII = Tripterygium glycosides.

4. It is interesting that the authors picked only seropositive RA as a comparison. Given that patients are seronegative with ICI-IA primarily, would have been interesting to compare to seronegative RA or spondyloarthritis.

Reply:

Thanks for the valuable suggestion. In this study, we aimed to investigate the difference between ICI-induced IA and traditional rheumatoid arthritis. Thus, seropositive RA were enrolled as control. We agree that the comparison between IA and seronegative RA would be interesting. According to the reviewer's suggestion, we additionally involved seronegative RA (our previously published scRNA-seq data)² as control and replicated our main findings. Firstly, *IL1B*^{hi} myeloid cells were significantly enriched in both PBMCs and SFMCs of the PD-1-IA patients, compared with seronegative RA (**Supplementary Fig. 3a-f**). Secondly, we observed a significant accumulation of synovial exhausted CD8⁺ T-cell population in the PD-1-IA patients, but not in the seronegative RA (**Supplementary Fig. 10**). We added these results in Supplementary Figs, which were in line with the main conclusions in the study (ICI-IA vs. seropositive RA).

Supplementary Fig. 3. a. Identification of 9 subclusters of myeloid cells across all SFMC samples. **b.** Quantification of the proportion of each myeloid cell subcluster in the SFMCs between IA_act and seronegative RA. **c.** The proportions of major cell types in SFMCs from all individuals in the IA_act and seronegative RA. **d.** Violin plots showing the marker genes expression of the myeloid subclusters in SFMCs. **e.** UMAP plot showing the *IL1B* gene expression of myeloid cells in PBMCs between IA_act and seronegative RA. **f.** Violin plot showing the *IL1B* gene expression of the macrophages in SFMCs between IA_act and seronegative RA.

Supplementary Fig. 10. a. Identification of 12 subclusters of T/NK cells across all SFMC samples (active IA and seronegative RA). **b.** Violin plots showing the expression of the marker genes of

T/NK subclusters in SFMCs. **c.** Quantification of the proportion of each T/NK cell subcluster in the SFMCs between IA_act and seronegative RA. **d.** The proportions of major T/NK cell types in SFMCs from all the individuals in the patient groups (IA_act and seronegative RA).

5. To fully understand the role of CD8+Tex cells (or any of the cell types for that matter), it would be useful to have PBMCs from other ICI treated patients who did NOT develop ICI-IA. Without that control it is hard to say whether they are relevant specifically to ICI-IA or just to ICI treatment.

Reply:

Thanks for this important point. According to the suggestion, utilizing the published scRNA-seq dataset³ of ICI-nonAE as control, we verified that the myeloid cells of PBMCs in the ICI-IA patients expressed significantly higher levels of *IL1B* compared with the ICI-nonAE group (**Supplementary Fig. 3g-k**). In addition, flow cytometry analysis validated the absence of enrichment of *IL1B*^{hi} myeloid cells in ICI-nonAE patients (**Fig. 2k-i**). Collectively, these results confirmed the major findings of our study.

Supplementary Fig. 3. g. Identification of 8 subclusters of myeloid cells across all PBMC samples. **h.** Quantification of the proportion of each myeloid cell subcluster in the PBMCs among the patient groups. **i.** Violin plots showing the marker genes expression of the myeloid subclusters in PBMCs. **j.** UMAP plots showing the *IL1B* gene expression of myeloid cells in PBMCs among the patient groups. **k.** Violin plot showing the *IL1B* gene expression of the CD14⁺ monocytes in PBMCs among the patient groups.

Fig 2. j. Quantification of IL1 β ⁺ CD14⁺ monocytes (percentage of IL1 β ⁺ cells in CD45⁺ CD14⁺ cells) among the patient groups (n=4 per group) by flow cytometry. **k.** Representative flow cytometry plots for (j).

References

1. Braaten TJ, *et al.* Immune checkpoint inhibitor-induced inflammatory arthritis persists after immunotherapy cessation. *Ann Rheum Dis* **79**, 332-338 (2020).
2. Wu X, *et al.* Single-cell sequencing of immune cells from anticitrullinated peptide antibody positive and negative rheumatoid arthritis. *Nat Commun* **12**, 4977 (2021).
3. Zhu H, *et al.* Identification of Pathogenic Immune Cell Subsets Associated With Checkpoint Inhibitor-Induced Myocarditis. *Circulation* **146**, 316-335 (2022).

Reviewer #2 (expert in single-cell RNA sequencing):

1. The major concern is that conclusions from receptor-ligand analysis are too speculative. The author should perform some functional analysis. Does CCL5 from activated CD8 T cells (anti-PD1 treated?) really induce chemotaxis of IL1B+ monocytes or macrophages? Does CXCL10 from IL1B+ monocytes or macrophages attract CD8+ T cells? Do IL1B+ macrophages induce T cell proliferation or activation?

Reply:

We appreciate the reviewer's reasonable concern. To validate if the cell-cell communication in IA is enhanced, we have conducted the functional analysis according to the reviewer's suggestion:

- 1) The *ex-vivo* Transwell experiment validated that the peripheral monocytes isolated from IA-act patients had increased migration under CCL5 compared with RA and HCs (**Fig. 6i**). Additionally, these monocytes from IA-act patients also had enhanced migration towards the CD8⁺ T cells, which can be attenuated at CCL5 blockade (**Fig. 6j-k**).
- 2) Similarly, we also observed a significant migration of CD8⁺ T cells under CXCL10 in IA-act patients compared with RA and HCs (**Supplementary Fig. 12i**).

Collectively, these results provide functional analysis supporting the findings from the scRNA-seq analysis that cell communications between CD8⁺ T cells and monocytes in IA-act were enhanced through CCL5 and CXCL10.

We have added the *ex-vivo* chemotaxis assay in Methods (**page 14-15, line 531-545, highlighted in yellow**) and Results (**page 8, line 303-307, 310-311, highlighted in yellow**).

Fig 6. i. Transwell migration of monocytes under the CCL5 treatment by the chemotaxis assay. The migrated cell ratio was the division of migrated cell counts with CCL5 treatment to those without CCL5 treatment. Comparisons were assessed by ratio paired-*t* tests within the groups, and unpaired two-sided Wilcoxon tests among the groups. **j.** Transwell migration of monocytes towards T cells among the patient groups (IA_act, HC and RA), comparing with unpaired two-sided Wilcoxon tests. **k.** Transwell migration of monocytes towards T cells in IA-act, with or without CCL5 blockade, comparing with a paired *t*-test. ns, nonsignificant.

Supplementary Fig. 12 i. Migration of T cells under CXCL10 treatment by the chemotaxis assay. The migrated cell ratio was division of the migrated cell counts with CXCL10 treatment to those without CXCL10 treatment. Comparisons were assessed by ratio paired t-test within each patient group or unpaired Mann-Whitney test among the patient groups. ns, non-significant.

2. To confirm T cell-macrophage interactions in SF of IA patients, imaging analysis may be helpful. Some subsets of CD8+ T cells may be heavily interacting with macrophages in SF of IA, but not in RA. Visualization of IL1B+ macrophages in SF using in situ hybridization is also interesting although it may be difficult.

Reply:

We appreciate the suggestion on using imaging analysis to visualize T cell-macrophage interactions. Given that the functional analysis suggested by the reviewer supporting our major findings, and the residual synovial fluid samples of the enrolled patients (5 milliliters per patient, almost used for single-cell RNA sequencing) are not enough to perform the *in-situ* hybridization on the SF smears, we hope that in our next study, we will collect the synovial membrane tissue and perform this additional imaging analysis.

3. In figure 1, the authors showed enrichment of T cells and reduction of myeloid cells in SF of IA_act. This is just an analysis of cell frequency. The authors should show the count of T cells and myeloid cells (possibly by flow cytometry?). If T cells are accumulated in SF of IA, the percentage of myeloid cells may go down. At the same time, the author can also validate the finding of figure 5 by flow cytometry (accumulation of some CD8+ T cell subsets)

Reply:

We appreciate the valuable suggestions. Accordingly, we have added the data in Supplementary Fig. 1c-d. The bar plot showed cell counts by single-cell sequencing, and flow cytometry validated the counts of some T cells and macrophage subsets in SFMC of IA_act and RA. We observed the accumulation of CD8+ T cells and decrease of macrophages in SFMC of IA_act.

Supplementary Fig. 1. c. The cell counts of subsets in SFMCs of IA_act and RA patients. **d.** Flow cytometry plots presenting T cells and macrophage subsets in SFMCs of IA_act and RA patients.

4. The authors used PMA and ionomycin to detect IL-1B, and CCL3, and CXCL10 in monocytes/macrophages. PMA and ionomycin are normally used to activate T cells. Have you detected these cytokines and chemokines in macrophages without these stimulations?

Reply:

Thanks for the insightful comment. PMA and ionomycin are normally activators of T cells¹, and they can also activate the monocytes and macrophages². In our study, we stimulated all of the mononuclear cells with PMA and ionomycin. Without these stimulations, we could not detect the secretion of cytokine/chemokines (IL-1 β , CCL3, and CXCL10) in macrophages by flow cytometry (Figure A).

Figure A. Flow cytometry of the myeloid cells in SFMCs without the stimulation of PMA and ionomycin.

References

1. Smith ME, van der Maesen K, Somera FP, Sobel RA. Effects of phorbol myristate acetate (PMA) on functions of macrophages and microglia in vitro. *Neurochem Res* **23**, 427-434 (1998).
2. Foey AD, Brennan FM. Conventional protein kinase C and atypical protein kinase C ζ differentially regulate macrophage production of tumour necrosis factor- α and interleukin-10. *Immunology* **112**, 44-53 (2004).

5. In figure 2, the authors showed IL1B was high in IL1B+ PB monocytes in IA-act but not those in IA_rem. What are the differences and similarities between them? DEG analysis of the subset between IA_act and IA_rem may be interesting.

Reply:

Thanks for the valuable comment. We conducted a DEG analysis within the IL1B^{hi} monocyte subsets in PBMC, comparing IA_act with IA_rem (**Supplementary Fig. 7a**). The top 10 DEGs identified in IL1B^{hi} subset were consistent with those identified in the overall monocytes. The GSEA results also mirrored the upregulated pathways in the overall monocyte population (**Supplementary Fig. 7b**).

Supplementary Fig. 7. a. DEGs of IL1B^{hi} monocyte subsets in PBMCs in IA_act compared with IA_rem. The top 10 and bottom 10 DE genes were labeled. **b.** GSEA showing the significantly altered inflammatory pathways of IL1B^{hi} monocytes in PBMCs in IA_act compared with IA_rem.

6. In figure 1a, do you need to mention about ELISA? If I understand correctly, the authors used ELISA just for validation of some cytokines and chemokines and those results are not included in figure 1.

Reply:

Thanks for the comment. We are sorry for the inexplicitness. We intended to present the whole study design in Figure 1a. In addition to single-cell sequencing, we also performed validations by ELISA and flow cytometry. Thus we have amended the flow cytometry and ELISA diagrams in **Figure 1a**.

We thank the reviewers again for their helpful comments.

REVIEWERS' COMMENTS

Reviewer #1 (Remarks to the Author):

The authors have adequately addressed the comments of both reviewers. This has improved the manuscript. This is an important topic for study with minimal translational data available. No further comments.

Reviewer #2 (Remarks to the Author):

The authors have adequately addressed my concerns in their revised manuscript.